# Views of the Pharmacists’ Role in HPV Vaccinations: A Qualitative Study in Tennessee

**DOI:** 10.3390/pharmacy12020057

**Published:** 2024-03-28

**Authors:** Alina Cernasev, Kenneth C. Hohmeier, Oluwafemifola Oyedeji, Kristina W. Kintziger, Tracy M. Hagemann

**Affiliations:** 1Department of Clinical Pharmacy and Translational Science, College of Pharmacy, University of Tennessee Health Science Center, 301 S. Perimeter Park Dr., Suite 220, Nashville, TN 37211, USA; khohmeie@uthsc.edu (K.C.H.); thageman@uthsc.edu (T.M.H.); 2Department of Public Health, University of Tennessee, Knoxville 390 HPER, 1914 Andy Holt Ave., Knoxville, TN 37996, USA; oonaade@vols.utk.edu; 3Department of Environmental, Agricultural & Occupational Health, College of Public Health, University of Nebraska Medical Center, Omaha, NE 68198, USA; kkintziger@unmc.edu

**Keywords:** pharmacy, pharmacist, Human Papillomavirus (HPV), vaccine, hesitancy

## Abstract

The Human Papillomavirus (HPV) is a frequently occurring sexually transmitted infection in adults and is associated with various cancers that can affect both males and females. Recently, the Advisory Committee on Immunization Practices (ACIP) expanded its recommendations for the HPV vaccine to include patients aged 27–45 years with shared clinical decision-making. A commonly reported obstacle to receiving the HPV vaccine among adults is a lack of healthcare provider recommendations. Considering the suboptimal HPV vaccine coverage figures and noting that the vast majority of hesitancy research has been conducted among children and adolescents, limited research is available on the adult perception of HPV vaccination in pharmacies. This study focuses on understanding adults’ opinions and perceptions regarding the role of pharmacists in the uptake of the HPV vaccine and awareness of its availability in the pharmacy setting. Methods: After receiving approval from the Institutional Review Board (IRB), the qualitative study was initiated using virtual focus groups (FGs). Concepts from the Transtheoretical Model, the Health Belief Model, and the Social Cognitive Theory guided the study design. The corpus of data was collected in 2021 and 2022 by two researchers, and a third party transcribed the FGs to avoid any biases. The data were analyzed using Braun and Clarke’s Thematic Analysis. Results: Out of 35 subjects that participated in six FGDs, most identified as female, with ages ranging from 18 to 45 years. The following four themes emerged: (1) HPV vaccine awareness; (2) stigmas leading to reduced education and vaccination rates; (3) education preferences; (4) follow-up in vaccination series reminders and preferences. Conclusion: Participants’ views of the HPV vaccine and the ability to receive the vaccine in a pharmacy are influenced by a myriad of factors. Common factors include improved awareness, preferences for educational modalities, avoiding stigmas associated with HPV vaccination, combating gender-focused biases, and preferences for the location of vaccination. These barriers provide opportunities for pharmacists to promote and enhance vaccine uptake.

## 1. Introduction

The Human Papillomavirus (HPV) is a frequently occurring sexually transmitted infection in adults that is associated with various cancers that can affect both males and females [1]. In the United States, the HPV vaccine series is recommended for adolescents aged 9–12 years and for individuals up to age 26 as part of the catch-up vaccination process [2]. Recently, the Advisory Committee on Immunization Practices (ACIP) expanded its recommendations for the HPV vaccine to include those aged 27–45 years with shared clinical decision-making [2]. Although vaccination rates among adolescents have been steadily increasing over the years, data from a recent report based on the 2022 National Immunization Survey-Teen (NIS-Teen) showed no increase in vaccination rates from the previous year (the first-time occurrence since 2013) [3]. HPV vaccination rates are below the desired goal for adolescents, with 76% of adolescents aged 13–17 years receiving one or more doses, and 62.6% of this group completed the series in 2022 [3,4]. Among adults, one report showed that the percentage of adults aged 18−26 years who completed the recommended doses of the HPV vaccine was 21.5% in 2018 [5].

A commonly reported barrier to receiving the HPV vaccine among adults is the lack of healthcare provider recommendations [6,7,8]. A plausible method to combat this issue could involve increasing public awareness of the availability of pharmacists and their ability to administer vaccines, including the HPV vaccine. Pharmacies are an appealing option for HPV vaccination due to their ease of accessibility and convenient hours [9]. Nearly 90% of the United States population has a community pharmacy located within five miles of their residence [10]. Additionally, patients visit their local pharmacy frequently, about twice as often as they report visiting their doctor or healthcare provider [11]. Addressing barriers and promoting pharmacies as HPV vaccination sites is important, particularly in rural areas with disproportionately lower HPV vaccination rates [12].

Studies among pharmacists have documented several barriers to HPV vaccination. Findings from a systematic review of the literature focused on pharmacists’ perceived barriers to administering the HPV vaccine showed multi-level impediments to vaccination [13]. Patient-level factors may include financial barriers, patient refusal, poor demand, safety concerns, lack of knowledge, or low education [13]. For adolescents, pharmacists reported several parental barriers, including safety and efficacy concerns, lack of education, lack of consent, cost issues, inadequate demand, stigma, hesitancy about having conversations around sexuality, beliefs that children are not at risk for HPV, beliefs that vaccination is not age-appropriate, and perceived approval of premarital sex or risky behavior following HPV vaccination [13]. Barriers related to pharmacists’ personal perceptions or attitudes include safety concerns, fear of potential adverse events, the sensitivity of HPV discussions, inadequate information, and misinformation. Multiple system- or organization-level barriers were also reported across studies [13]. These included challenges with reimbursement, compensation, insurance coverage, lack of time due to competing priorities, difficulty screening and tracking of patients, cost of purchasing and stocking vaccine products, lack of educational materials for patients, provider competition, liability concerns, inadequate space, and staffing issues [13].

Several studies among parents or caregivers have documented varying levels of acceptance of pharmacies as vaccination sites [14,15,16,17]. Parental or caregiver attitudes and perceptions may influence acceptance of HPV vaccine uptake in pharmacies, including knowledge of pharmacists’ training and competency in providing vaccination services [14,15], concerns about the transfer of vaccination records to their doctor’s office [15,16], concerns about privacy in pharmacy settings, staffing changes, and the absence of an established patient–pharmacist relationship [15].

While previous studies have largely focused on adolescents, limited research is available on the adult view of HPV vaccination in pharmacies. Understanding the role of pharmacies in HPV vaccine uptake is an important step in identifying barriers and improving HPV catch-up vaccination among adults. Additionally, adults’ perceptions of the HPV vaccine can be leveraged by pharmacies that serve as an accessible, alternative vaccination site for adults. Therefore, this study focuses on understanding adults’ opinions and perceptions regarding the role of pharmacists in the uptake of the HPV vaccine and awareness of the vaccine’s availability in the pharmacy setting.

## 2. Methods

A qualitative focus group (FG) methodology was employed to understand and characterize Tennessee residents’ perceptions about HPV vaccination programs offered through pharmacies to improve access to quality care [18]. Subjects for this study were recruited via fliers placed in various venues throughout Tennessee to capture rural and urban representation. Subjects self-selected their participation, and the inclusion criteria were as follows: (1) adults 18–45 years old, (2) residing in the state of Tennessee, (3) English speakers, (4) familiarity with the topic, and (5) willingness to discuss their opinion. Subjects received a gift card as compensation for their participation. The University of Tennessee Institutional Review Board approved this study (IRB # 21-08416-XM, approved 3 November 2021).

The FG guide was developed based on the Transtheoretical Model, the Health Belief Model, and the Social Cognitive Theory for constructs to examine participant perceptions and experiences regarding the opportunity to receive the HPV vaccine from a pharmacy [19,20,21]. The FG comprised open-ended questions that probed views on viable solutions to overcome the barriers of receiving an HPV vaccine from the pharmacy. All FGs were conducted using an online platform by two researchers. Verbal informed consent was obtained prior to the FG, in which the procedures were explained to the subjects and any questions about participation were answered [18]. All the subjects agreed to be audio recorded, and verbatim transcription was manually conducted by an objective professional company to minimize bias. The data collection occurred from November 2021 to April 2022, when thematic saturation was achieved [22]. More information on data collection has been published in a previous manuscript [23].

Themes and sub-themes were identified using thematic analysis with an inductive approach described by Braun and Clarke [22]. Data analysis followed the six-step process for conducting a thorough and transparent data analysis [22]. After familiarization with the data, the transcripts were coded with verbatim pieces of text, and then preliminary codes were generated; similar codes were grouped into categories [22]. All the categories were clustered and analyzed to reveal the major themes. Two researchers performed a line-by-line reading of the first two FG transcripts after familiarizing themselves with the corpus of data [22]. After review, the researchers’ team collaborated to refine and define the initial codes as well as identify emergent codes. The coding process was facilitated by qualitative software, Dedoose^®^ (v2.0, Manhattan Beach, CA, USA), which was used for generating initial codes and developing and reviewing themes. This study is reported in accordance with the consolidated criteria for reporting qualitative research (COREQ) [24].

## 3. Results

Out of 35 subjects that participated in six FGs, most identified as female, with ages ranging from 18 to 45 years.

Four major themes were identified from the thematic analysis, which included the following: (1) HPV vaccine awareness; (2) stigmas leading to reduced education and vaccination rates; (3) education preferences; and (4) follow-up in vaccination series reminders and preferences. Table 1 presents the themes and sub-themes.

### 3.1. Theme 1: HPV Vaccine Awareness

Due to the focus of this study, which sought to provide insight on adults’ opinions and preferences, one of the primary goals was to understand if the adults involved within the group were aware of vaccination availability, how they were made aware, and gauge if they understood the age groups the vaccination is indicated for.

One patient was able to reflect on their memory of the “old” Gardasil® commercial. The participant said the following:

“I can’t remember when it was, I feel like it was in the early to mid-2000s, I remember that there was a really catchy Gardasil commercial that was on TV, and that’s actually the first place that I remember seeing it…I just remember it going through my head constantly, and I was probably in like elementary school when this happened, so that was a long time ago.” (FG3)

Although this memory is from over 10 years ago, it is unknown if the persons in this FG are aware of the recommendation to receive the HPV vaccine until the age of 45 with shared clinical decision-making. Furthermore, the bottom-line recommendation for the HPV vaccine is to initiate the series in males and females beginning at age 9–25, with catch-up vaccines available to those who did not complete the doses in childhood.

Another focus group participant admitted the following:

“I did not know that you could get it up to age 45, which I’m kind of right there at the cut-off, 44. I think it’s something I would probably like to research a little bit more on for myself and talk to my doctor about it to get some more information.” (FG2)

This patient still seemed to find benefit in speaking to their doctor about the HPV vaccination and became aware that they should seek additional information.

Looking at awareness and information regarding the vaccine, the same patient mentioned a lack of educational material provided at the pharmacy. The participant said the following:

“I’ve never seen stuff about HPV at my pharmacy, and honestly, I really would like to see it. I don’t even know if they would offer that at a pharmacy, so seeing it at a pharmacy would be-- I think would be beneficial to me…” (FG4)

### 3.2. Theme 2: Education Preferences

It is clear that enhanced awareness needs to be conjured within communities to better explain the benefits of HPV vaccination to patients eligible to receive the series. Another goal arose to better understand the preferences that these patients within the groups had for the dissemination of education regarding HPV vaccination. One participant mentioned wanting to understand the relevance of vaccination.

“I want a pamphlet, but in that pamphlet, I want it to have statistics. I like numbers. So, I want it to show me how it affects people of color, since I’m a person of color. So, if it’s 60% of African American women get or die from HPV-- I want to know that… I want to know the numbers. That puts it into perspective for me and makes it more relevant, I should say.” (FG4)

This sentiment was echoed by another focus group member, but new emphasis was placed on understanding side effects. Overall, understanding education preferences is important to ensure that the materials created and disseminated can address the concerns most pertinent to patients. Another participant mentioned the following:

“When it comes to the HPV vaccination, I would want to know, because when it comes medicine, medicines always have-- you know, it helps this-- it prevents this, but it causes this.”(FG1)

### 3.3. Theme 3: Stigmas Leading to Reduced Education and Vaccination Rates

Recognition of education preferences is necessary; however, due to the stigmas that were mentioned, it is also important to use education as a tool to address these stigmas. One participant commented the following:

“When I first heard about it (HPV vaccination) … it wasn’t presented as a sexually transmitted disease, and I still don’t feel like it is being presented like that. Some might present it that way, but the way that it was presented to me was that it could just happen…I feel like that part of it is confusing within itself because how can I get it? I need to know how I can prevent myself from getting HPV. If it’s just walking past somebody, touching them, kissing them, I need to know the specifics.” (FG4)

Additionally, a focus group member brought to light that there were some parents who worried that allowing their children to be vaccinated for HPV would permit promiscuity. They explained the following:

“I think STD stigma is such a huge part of it. There was a woman who was a teacher at my high school who taught anatomy and who was a breast cancer survivor herself, and yet she had a huge thing about how it was really hard for her to get the HPV vaccine for her teenage daughter because she worried that she would consider it permission to have sex, which even at that age, like baffled me.” (FG2)

Another member of the focus group who could speak to her own experience with the HPV vaccination mentioned the following:

“When it (the HPV vaccine) was introduced to me, I can remember I was having a checkup at my pediatrician’s office, I think I was 14, and my mother was there, and I was not sexually active, and it was a very uncomfortable conversation for my mom. She was extremely offended by our pediatrician introducing the idea of me having the HPV vaccination. So, I think there was definitely a level of misunderstanding.” (FG2)

Despite being in the presence of a healthcare provider, there was still discomfort discussing the benefits of a preventative vaccination. Another participant spoke about a possible remedy to the stigma surrounding vaccination, starting with the following:

“I think another stigma with the HPV vaccine is you can only get it mainly like from having sex, right? So, a lot of moms and dads are like, oh, I don’t want my 16-year-old to get the HPV vaccine, she’s not having sex or anything, right? So probably just saying like it’s not-- I think it’s like kind of stigmatized because … they don’t think that their kids are doing it, so probably just talking about that in like a more positive way, just things like that.” (FG4)

The need for clinicians to speak confidently and provide education to dispel stigmas is apparent.

As generational changes occur, the need for continued education is emphasized. This idea was well represented by one contributor, who said the following:

“It’s because there are more-- the millennials-- what’s a better term-- they’re-- it’s more open, sex is a more open topic and more open about doing it than it may have been in our years or whatever before then. But now that it’s so open, this is something that we need to have a discussion about. Everybody is having sex. The age of children having sex is becoming younger and younger, so, yeah, that conversation is prevalent and needed.” (FG5)

### 3.4. Sub-Theme: Gender Biases in the Need for HPV Vaccination and Education Surrounding HPV

Another area that educational approaches can concentrate on is addressing gender bias within HPV vaccination efforts. One sharer mentioned that education should describe benefits for both males and females. The following was said:

“I would want to know what the HPV vaccination did, and what it did for men, what it did for women, you know, what are the benefits, what are the risks of the vaccination.”(FG6)

Another participant echoed this same idea, but given his new understanding of vaccination, he reflected by saying the following:

“Knowing how it (a male getting vaccinated for HPV) affects women, I would also get it because it would benefit both of us, you know, if you’re dating or something or your significant other, it’s a win-win situation.” (FG5)

The blindness regarding the benefits of HPV vaccination among both males and females was well summarized by a contributor, who stated the following:

“There is definitely a lack of knowledge on HPV, as you heard the men on this call didn’t know much about it, but that pretty much represents a lot of men. They don’t know about it, or they think that it doesn’t affect them. It could be because they don’t go to the doctor a lot. It could be because their doctor is not looking for it because, from my understanding, it’s a painful way to find out. But not knowing is not a good-- it’s not a good reason to not want to find out about it or-- you know, they should know because it affects them as well. It doesn’t just affect women. They just don’t know that they have it. And so, I think education should be steered towards as men as well because it’s not really gender specific, but it seems like more girls and more women are educated on it than boys and men.” (FG1)

The importance of gender-inclusive educational materials was highlighted in all focus groups, bringing to light the consistent need for updated guidance that reduces stigmas and addresses these concerns.

### 3.5. Sub-Theme: Location Preferences

In addition to education preferences, location preferences were discussed by all groups, and the convenience of receiving vaccinations at local pharmacies was also mentioned.

“I go to [pharmacies] more often than I go to the doctor. You know, like I have to pick up prescriptions and stuff like that, so I go there more often than I go to see my doctor, so I think it would be better to get it there than it would be to get at my doctor. And also, a lot of time-- well, now, with the doctor, it’s hard to get an appointment, and it’s easier to get an appointment at like a pharmacy because they’re open 24/7, you don’t really have to make an appointment, and a lot of their stuff is walk-in.” (FG4)

At another point, when asked if pharmacies should be proactive in providing educational materials, a member commented the following:

“I don’t think that I would be bothered by that. They offer you the flu shot when you go in to pick up your medicine, you know, so why not this as well?” (FG4)

When further discussion surrounding pharmacies arose in other groups, the topic of follow-up after providing educational materials was mentioned. One member emphasized the preference to do their own diligence before receiving vaccination. This was reflected in the following quote:

“I wouldn’t necessarily say that would be the best idea to get it right then and there because there might be more information you’d like to gather on your own or think about it. But having the option to get it then, if that’s something you’re into, I guess would be a good idea, but also not pushing people to get it at the exact same time they hear about it for the first time.” (FG6)

Additional members voiced their preference for time between the initial presentation of educational materials by saying the following:

“I agree (with another participant) that you shouldn’t be pushed to get it that day, but you should have the option to once you understand if you want it or not.” (FG6)

In contrast to receiving vaccinations at a pharmacy, some participants mentioned that their preference was to receive vaccinations at the doctor’s office instead of the pharmacy. They spoke about this preference by explaining the following:

“I think only because a lot of the pharmacies I’ve been to, they just put up like a janky screen, and it doesn’t feel super clean. I’m sure it is. And it happened at one where there was a room, and it was really nice and lovely, but the person that was writing out the cards, like her hands were like shaking, and I was like, this person is about to jab me with a needle, and it like made me a little bit nervous.” (FG2)

A focus group member who disclosed their personal struggle with HPV explained the following:

“Sometimes it is just uncomfortable to have those types of conversations with medical professionals, but a person may receive the message better from someone like you. You know, you would have more influence on individuals because you know you can speak to them on a level that they can maybe better understand and receive the message well. So, I absolutely think using individuals within the community to inform individuals about the HPV vaccination would definitely be a positive influence on individuals receiving it.” (FG2)

### 3.6. Theme 4: Follow-Up in Vaccination Series Reminders and Preferences

Vaccination follow-up was another area of interest discussed. When discussing additional reminders for vaccination follow-up doses, one member mentioned the following:

“As far as the reminder for the other two doses, I think a text message or I don’t even know how long it is between doses, but if the next doses happen to be at the same time that you’re picking up your next prescription, you know, the next refill of your prescription, then that would be an easy way to do it.” (FG6)

A participant from the same focus group mentioned the need for reminders and said they were likely to forget if they did not schedule an appointment in advance, saying the following:

“I’m not sure if I’ll remember to go back and schedule in a timely manner, so having either the option to schedule another appointment (for follow up) right then and there or get reminders to schedule another appointment would be beneficial for me.” (FG6)

Follow-up was also discussed in the context of post-vaccination side effects, with one participant mentioning the following:

“If I’m given a vaccine at my doctor’s office, I would like my doctor’s office to follow up. I think if I got it at a pharmacy, I would like the pharmacist to follow up. Regardless of whoever gave me the vaccine, I would like them to follow up with me to see if I have any symptoms.” (FG4)

This sentiment was echoed elsewhere, with a participant stating the following:

“I think there should be an immediate follow up maybe a week after, like, hey, how are you feeling, is there anything additional that we can talk about? You know, are there any questions that you have that are unanswered so that individuals do feel comfortable coming back.” (FG2)

Follow-up was also mentioned as a strategy to build additional trust and rapport with healthcare professionals in vaccination efforts, especially between patients and their pharmacists, as previous participants mentioned the lack of relationship with the pharmacist as a barrier to initiating the HPV vaccine series altogether. Using follow-up to check in, provide additional educational resources, and build patient rapport was discussed through almost all of the focus group sessions.

## 4. Discussion

This study explored adults’ perceptions of HPV vaccination as it relates to pharmacy-based vaccine promotion, education, screening, and administration. This population is in contrast with the overwhelming amount of perception research related to HPV vaccination in children [25,26,27,28]. Despite the uniqueness of the population of interest, there were several commonalities between pediatric and adult populations when it came to barriers to receiving the HPV vaccine, including stigma, awareness, and the quality of provider recommendations [28]. However, there were also distinct differences, including a lack of awareness of the ACIP recommendation for the vaccine among the adult population aged 27–45, a desire for a greater level of detail on the safety and efficacy of the vaccine, a need for convenient locations, reminders to facilitate vaccine series completion, and a desire on behalf of male participants to become vaccinated, even if they themselves are less likely to realize the benefits of the vaccine.

Generally, participants were aware of the HPV vaccine, although specific nuances were less well known, including the ACIP recommendation for the vaccine for persons aged 27–45, the ability for males to receive the vaccine, and the ability to receive the vaccine in a pharmacy setting. Television commercials, social media, family, peer conversations, and provider recommendations were all listed as the primary means by which participants were made aware of the HPV vaccine. Despite general positive feelings and overall awareness about the vaccine, not all participants were vaccinated for reasons similar to those published in pediatric populations [28]. However, some unique findings in this population included “forgetfulness” to receive the vaccine, despite the intention to become vaccinated. Given that this adult population does not have the same routine HPV vaccine recommendations that a pediatric population may receive, a novel barrier is introduced in that patients may desire to be vaccinated, but given competing priorities in their lives and health, they may forget to request the vaccine. Furthermore, it was also felt that written, quantified details of the risks and benefits of the vaccine specific to the patient’s sex, race, and ethnicity would be preferred to help facilitate the decision to pursue vaccination. This also mirrors effective educational interventions seen in pediatric populations, suggesting that similar educational interventions may be successful in this population and warrant further investigation [29].

The group noted several misconceptions about the HPV vaccine. This included the public perception of the vaccine as an “STD vaccine” or that it promotes promiscuity. In particular, these perceptions were noted to be a barrier for participants receiving the vaccine within the pediatric vaccination age window and are well-known to be barriers to HPV vaccine uptake among other groups [30]. Of note, research has confirmed that receiving the HPV vaccine does not increase the likelihood of risky sexual behaviors or reduce the age of sexual debut [31,32]. It was also noted that the vaccine had become “gendered” as a female vaccine, and providing education to men eligible for the vaccine may boost vaccination acceptance among males, similar to what has been suggested in other studies [33]. Of note, male participants expressed a desire to become vaccinated to support overall community health, even if they did not realize a specific benefit.

When it comes to pharmacies serving the community as a setting for HPV vaccine administration, the pharmacy was regarded as a convenient location for receiving the vaccine and the educational materials that improve public health. However, it was also noted that there was variation in the aesthetics of the pharmacy itself and that some patients may decline vaccine services from those pharmacies that did not have a room that resembled a clinic setting to administer vaccines. This sentiment has been echoed in other research related to novel clinical services in pharmacy settings, like test-to-treat infectious disease care [34].

Given the results of this study, there are several areas where community pharmacy may be leveraged to improve public health through HPV vaccination. First, participants noted that the time constraints and competing priorities of their schedules led them to frequently forget to seek vaccination, even if they were willing to receive it. Past studies in pharmacy have demonstrated clearly that technology-enabled interventions that either prompt the pharmacist to have a conversation with the patient or message the patient directly on their phone are effective in prompting vaccinations [35,36,37]. Given that most patients visit a pharmacy almost twice as often as a primary care provider [38], the numerous touchpoints with the patient represent opportunities for reminding those patients who are already predisposed to accepting a vaccine recommendation. Similarly, several patients were unaware that the pharmacy offered the HPV vaccine, noting that they had never seen any promotion or recommendations around HPV vaccination. Therefore, interventions aimed at boosting screening with subsequent evidence-based recommendation approaches for HPV vaccination may also represent a potential future direction for this research, as similar approaches have been beneficial in this setting for other vaccines [39].

## 5. Conclusions

Participants’ views of the HPV vaccine and their ability to receive the vaccine in a pharmacy are influenced by a myriad of factors. Common factors include improved awareness, preferences for educational modalities, avoiding stigmas associated with HPV vaccination, combating gender-focused biases, and preferences for the location of vaccination. Several vaccine promotion strategies were perceived to positively influence HPV vaccine-seeking behaviors, which provide opportunities for pharmacists to promote and enhance vaccine uptake in the community.

## Figures and Tables

**Table 1 pharmacy-12-00057-t001:** Emergent themes and sub-themes.

Theme	Sub-Theme
Theme 1: HPV vaccine awareness	
Theme 2: Education preferences	
Theme 3: Stigmas leading to reduced education and vaccination rates	Sub-theme 1: Gender biases in the need for HPV vaccination and education surrounding HPV Sub-theme 2: Location preferences
Theme 4: Follow-up in vaccination series reminders and preferences	

## Data Availability

The data presented in this study are available in the article.

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
