# Peer review of "Views of the Pharmacists’ Role in HPV Vaccinations: A Qualitative Study in Tennessee"

_pharmacy, 2024, doi:10.3390/pharmacy12020057_

Round 1
Reviewer 1 Report
Comments and Suggestions for Authors
Authors have done a commendable job in presenting the data from the studies conducted among focus groups to understand the role of pharmacies and pharmacists in HPV vaccination awareness of general public. The article is well written. However, the study is conducted on a small sample size of 35 people and limited to one state during a short period of time. And it is more like text-based and hard to confirm. There is no graphs or tables to add to the study. The manuscript is written similarly to the qualitative study conducted by the authors on perspectives of HPV vaccination in adults.
Question: Was there a pharmacist in the sample size selected who could give more insight to the HPV vaccination and if yes, it is worth mentioning in the manuscript.
No major comments other than typos that needs to be addressed.
Author Response
Question: Was there a pharmacist in the sample size selected who could give more insight to the HPV vaccination and if yes, it is worth mentioning in the manuscript.
Response: Thank you for this suggestion. The sample size of this qualitative study was dependent on thematic saturation, which was explained in lines 115-116.
No major comments other than typos that needs to be addressed.
Reviewer 2 Report
Comments and Suggestions for Authors
I read with interest the paper titled "Views of the Pharmacists’ Role in HPV vaccinations: A qualitative study in Tennessee"
- A good background was provided. My main question is related about why do you focus only on pharamcists as the type of professionals that are in position to administer vaccination in pharmacies? From a quick search, it appear also pharmacy technicians are acting in vaccine administration in different parts of US. Please use those references to add in the manuscript, and give a broad vision of the role of pharmacies, with both types of professionals acting as vaccinators. I suggest the discussion of the topic.
https://www.mdpi.com/2226-4787/7/4/152 ("pharmacy technicians trained and certified to administer immunizations increase access to vaccination care and have the potential to drastically increase the number of immunizations")
https://www.sciencedirect.com/science/article/abs/pii/S1544319118300049 ("pharmacy technicians have legally administered immunizations in the United States. Trained pharmacy technicians demonstrated knowledge of vaccination procedures and self-reported improved confidence in immunization skills")
https://www.sciencedirect.com/science/article/abs/pii/S1544319123002492 ("The utilization of pharmacy technicians in administration helped to accelerate the immunization process, alleviate the burden on pharmacists and other health care professionals, and ensure widespread vaccine distribution")
https://www.mdpi.com/2076-393X/10/8/1354 (Multiple states are enacting legislation to include technician vaccine administration as a permanent component of their scope of practice.)
https://www.sciencedirect.com/science/article/abs/pii/S1544319121000054 (pharmacy patients support the additional role of pharmacy technicians as immunizers in general.)
What does "Subjects self-selected their participation" mean?
- How were participants selected for the focus groups?
- Transcripts were made manually or software-helped? If so, please add the information on the software used.
- I suggest presenting the main data for the themes and subthemes in a table. Its hard to read and find the main results this way.
Author Response
Response: Thank you for suggesting expanding on the topic of pharmacy technicians administering vaccines. However, the scope of this manuscript and the data collection was focused on pharmacists. Additionally, the services that pharmacy technicians are providing are made available to them under the direct supervision of a registered pharmacist. Pharmacy technicians can administer vaccines once they obtain the appropriate license/ credentials. Furthermore, by the federal and state laws, the pharmacy technicians cannot recommend and initiate a vaccine as it is the pharmacist’s scope of practice.
What does "Subjects self-selected their participation" mean?
Response: Thank you for this clarification. After seeing the ads, the participants scanned a QR code and enrolled in the study.
- How were participants selected for the focus groups?
Response: Thank you for this suggestion. The participants were selected after screening for the inclusion criteria.
- Transcripts were made manually or software-helped? If so, please add the information on the software used.
Response: Thank you for this inquiry. The transcripts were manually transcribed. We amended the manuscript.
- I suggest presenting the main data for the themes and subthemes in a table. Its hard to read and find the main results this way.
Response: Thank you for the valuable suggestion. We created a table and amended the manuscript.
|
Theme |
Sub-theme |
|
Theme 1: HPV vaccine awareness |
|
|
Theme 2: Education Preferences: |
|
|
Theme 3: Stigmas leading to reduced education and vaccination rates |
Sub-theme 1: Gender biases in needs for HPV vaccination and education surrounding HPV Sub-theme 2: Location Preferences
|
|
Theme 4: Follow up in vaccination series reminder and preferences |
|